# Influence of Seasonings and Spice Essential Oils on Acrylamide Production in a Low Moisture Model System

**DOI:** 10.3390/foods11243967

**Published:** 2022-12-08

**Authors:** Yuchen Zhu, Bobo An, Yinghua Luo, Xiaosong Hu, Fang Chen

**Affiliations:** College of Food Science and Nutritional Engineering, National Engineering Research Centre for Fruits and Vegetables Processing, Key Laboratory of Storage and Processing of Fruits and Vegetables, Ministry of Agriculture, Engineering Research Centre for Fruits and Vegetables Processing, Ministry of Education, China Agricultural University, Beijing 100083, China

**Keywords:** acrylamide, seasonings, spice essential oils, Maillard reaction, low moisture model system

## Abstract

Acrylamide (AA) is a typical contaminant produced during the heating process. In the present study, two seasonings (soy sauce and rice vinegar) and three spice essential oils (chive, ginger, and pepper) were added to the asparagine (Asn)/glucose (Glc) diethylene glycol model system to investigate the production of AA in a low moisture model system. The generation of AA was significantly enhanced when low levels of soy sauce (1% and 3% *v*/*v*) were added (*p* < 0.05). The Asn/Glc model system was heated for 15 min with 0%, 1%, or 3% (*v*/*v*) soy sauce, containing 43 mg/L, 63 mg/L, and 53 mg/L AA, respectively. However, the addition of a high level of soy sauce (5% *v*/*v*) showed significant inhibition of AA production after heating for 10 min (*p* < 0.05). About 36% of AA was inhibited in the Asn/Glc/soy sauce (5%) model system after heating for 15 min. The addition of low levels of rice vinegar (1% and 3% *v*/*v*) showed comprehensive effects on AA production. Nevertheless, the addition of rice vinegar at 5% *v*/*v* had an inhibitory effect on AA generation (*p* < 0.05). All kinds of spice essential oils promoted the production of AA (*p* < 0.05). There was a dose–response relationship between the level of spice essential oils and the generation of AA. This study proposes the importance of seasonings and spice essential oils for AA production in food preparation.

## 1. Introduction

The Maillard reaction plays an essential role in the food industry as it contributes to the browning, texture, flavor, and aroma of food [1]. However, the Maillard reaction can generate some neo-formed contaminants which may pose health risks to humans [2]. Acrylamide (AA) is a typical contaminant produced in carbohydrate-rich foods when the thermal temperature is higher than 120 °C [3,4]. Reducing sugars and asparagine (Asn) are reported as the main precursors of AA [5]. AA has different types of toxicity such as neurotoxicity, genotoxicity, and reproductive toxicity [6]. The International Agency for Research on Cancer has classified AA as a Group 2A carcinogen [7].

Numerous studies have been conducted to measure the AA levels in various foods and to evaluate the intake of AA for the general population [8,9,10]. High concentrations of AA have been found in potato products (5–8440 μg/kg), crackers (5–2110 μg/kg), cookies and granola bars (5–1796 μg/kg), breakfast cereals (5–1354 μg/kg), and coffees (5–1080 μg/kg), while low levels of AA have been detected in soft bread (5–102 μg/kg) [11]. Breakfast cereals, French fries, potato chips, cookies, and crackers have been reported to be the main food contributors of dietary AA in Western countries [11,12]. However, the contributors to the dietary source of AA are different in Asian countries. The greatest contributors to AA intake have been “vegetables and vegetable products” (35.2%), “cereals and cereal products” (34.3%), and “potatoes and potato products” (15.7%) according to data from the 5th Chinese Total Diet Study [13]. Particularly, stir-fried vegetables have been reported as the main contributor among “vegetables and vegetable products” due to stir-frying being one of the major cooking techniques for Chinese food processing [14]. Similarly, “vegetables” (11%) are also a major contributor to AA exposure in Japan [15].

However, few studies focus on the generation of AA in prepared dishes. Nowadays, the degree of dishes’ industrialization is continuously enhanced. Thus, more attention should be paid to the production of AA in prepared dishes. AA levels are directly affected by the type of food, the content of precursors in the raw material, and thermal conditions such as temperature, time, and humidity [16]. In addition, the seasonings and spices used in the cooking process also affect AA generation.

Soy sauce and rice vinegar are fermented seasonings that are widely used in Asian dishes. Chive, ginger, and pepper are important spices used in Asian dishes. These seasonings and spices are indispensable for the flavor and aroma of dishes. Nevertheless, few studies have investigated the influence of these seasonings and spices on AA production. This study aims to investigate the effect of seasonings and spices on AA generation. A diethylene glycol model system was prepared to model the low moisture conditions of cooking. Different levels of seasonings (soy sauce and rice vinegar) and spice essential oils (chive, ginger, and pepper) were added to the diethylene glycol model system and the generation of AA was measured.

## 2. Materials and Methods

### 2.1. Reagents and Chemicals

AA, Asn, and glucose (Glc) (>99%) were obtained from BioDee Biotechnology Co., Ltd. (Beijing, China). Diethylene glycol was ordered from Macklin Biochemical Co., Ltd. (Shanghai, China). Labeled ^13^C_3_-AA (isotopic purity 99%) was bought from the Cambridge Isotope Laboratories (Andover, MA, USA). LC solvents were ordered from Honeywell (Seoul, Republic of Korea). Soy sauce, rice vinegar, chive essential oil (CEO), ginger essential oil (GEO), and pepper essential oil (PEO) were purchased from the local market.

### 2.2. Model System Preparation

The Asn/Glc model system consisted of Asn (0.01 g) and Glc (0.018 g). Both reactants were thoroughly mixed and dissolved by mixing diethylene glycol and deionized water to obtain three water activities (a_w_) of 0.173, 0.270, and 0.442 (Table 1). The solutions were sealed in 40 mL glass tubes and heated under 180 °C for 36 min by using a parallel synthesizer (Radleys, UK). Samples were collected every 4 min. Then, the tubes were immersed in an ice bath to avoid further reactions.

### 2.3. Effect of Seasonings and Spice Essential Oils on AA Production

Different amounts of soy sauce (1%, 3%, or 5% *v*/*v*), rice vinegar (1%, 3%, or 5% *v*/*v*), CEO (0.5%, 1%, or 3% *v*/*v*), GEO (0.5%, 1%, or 3% *v*/*v*), and PEO (0.5%, 1%, or 3% *v*/*v*) were added to the Asn/Glc model system (a_w_ 0.27), respectively. Samples were collected every 5 min. Subsequently, the tubes were immersed in the ice bath.

### 2.4. Determination of AA

The ^13^C_3_-AA solution (120 mg/L, 10 μL) was added to the sample (1 mL). The quantification of AA was performed by an ultra-performance liquid chromatograph-tandem mass spectrometer (UPLC–MS/MS) according to the method described in our previous study [17].

### 2.5. Determination of a_w_

The a_w_ was measured with a water activity meter (Novasina, Zurich, Switzerland).

### 2.6. Statistical Analyses

All experiments were performed in triplicates and the data are presented as mean ± standard deviation (SD). The statistical analysis was conducted with SPSS Statistics V22.0 (IBM Corporation, Armonk, NY, USA), and *p* < 0.05 was significant. Graphs were drawn using OriginPro 9.1 software (OriginLab, Northampton, MA, USA).

## 3. Results and Discussion

### 3.1. Effect of a_w_ on the Production of AA

The level of a_w_ plays a critical role in the production of AA [18,19,20]. The food surface was at a low water content during the cooking process [21]. Therefore, a low a_w_ model system was more suitable when it was compared to an aqueous model system (a_w_ 1). Since diethylene glycol has a unique boiling point and can be sufficiently mixed with water, diethylene glycol was chosen to prepare a low a_w_ model system. Three different a_w_ (0.173 ≤ a_w_ ≤ 0.442) were studied. For all experimental a_w_ conditions, the concentrations of AA initially increased with increasing heat time from 4 to 28 min, and subsequently, decreased with time (Figure 1). Additionally, the decreased a_w_ resulted in an increase in AA concentration (*p* < 0.05). The maximum concentrations of AA were 38 mg/L (a_w_ 0.442), 46 mg/L (a_w_ 0.270), and 56 mg/L (a_w_ 0.173) heated at 180 °C for 28 min, respectively. These results were in agreement with a previous study that showed that the generation of AA increased with the decrease of a_w_ [21]. In previous studies, the powders of precursors were mixed to prepare the low-moisture model system [22,23]. However, this method induces the carbonization of powers and the production of insoluble substances, which hinders the extraction and detection of target substances. Compared to the powder model system, the production of insoluble substances was avoided in the diethylene glycol model system. Therefore, diethylene glycol is more appropriate for the preparation of a low-moisture model system.

The model system at a_w_ 0.270 is better than a_w_ 0.442 and 0.173 for investigating the production of AA. The generation of AA was limited at a_w_ 0.442. On the contrary, the generation and elimination of AA were both enhanced at a_w_ 0.173, which resulted in a rapid change in AA concentration. This change was a disadvantage of studying the effect of seasonings and spice essential oils on AA production. Therefore, a_w_ 0.27 was selected for further experiments.

### 3.2. Effect of Seasonings on the Production of AA

Soy sauce is a fermented soy product, which is widely used in Asian dishes. Soy sauce contains reducing sugars, amino acids, peptides, fatty acids, aldehydes, Na^+^, and antioxidants [24]. The AA concentration significantly increased when 1% or 3% (*v*/*v*) soy sauce was added to the Asn/Glc model systems (Figure 2A) (*p* < 0.05). However, soy sauce demonstrated inhibitory effects on AA when the additional amount of soy sauce was 5% (*v*/*v*) (*p* < 0.05). On the one hand, the extra reducing sugars, Asn, and aldehydes provided by soy sauce could enhance the production of AA. Nevertheless, amino acids (except for Asn), peptides, Na^+^, and antioxidants provided by soy sauce could inhibit the production of AA [25,26,27,28].

Rice vinegar is a lightly sweet vinegar derived from fermented rice, primarily popularized in Asian countries. Chinese food standards indicate that the total acid content of rice vinegar must not be less than 3.5 g/100 mL [29]. Acetic acid is the major organic acid in rice vinegar. In addition, rice vinegar contains many other compounds such as reducing sugars, alcohols, aldehydes, and polyphenols [30,31,32]. In the initial stage of the reaction (0–10 min), the addition of 1% (*v*/*v*) rice vinegar significantly increased the AA concentration when compared to the Asn/Glc model systems (Figure 2B) (*p* < 0.05). Subsequently, the concentration of AA between the Asn/Glc and the Asn/Glc/rice vinegar (1%) model systems showed no significant difference (15–25 min). The fermented products of rice vinegar such as reducing sugars and aldehydes were involved in the production of AA, which increased AA concentration [33]. After heating for 15 min, an inhibition of AA was observed in both the Asn/Glc/rice vinegar 3% (*v*/*v*) and Asn/Glc/rice vinegar 5% (*v*/*v*) model systems (*p* < 0.05). These results may be attributed to the acetic acid in rice vinegar which has been reported to inhibit the production of AA [34,35]. Jung and colleagues proposed that lowering the pH value converted the free unprotonated amine of Asn to protonated amine, which effectively blocked the reaction between Asn and Glc and thus inhibited the generation of AA [36]. In addition, the phenols and flavonoids in rice vinegar may also contribute to AA inhibition [32].

### 3.3. Effect of Spice Essential Oils on the Production of AA

Chive (*Allium schoenoprasum* L.) is a perennial plant whose slender leaves are used as a spice. Chive contains many chemical constituents such as aldehydes, sterols, polyphenols, and flavonoids [37]. CEO was added to the Asn/Glc model systems to investigate the effect of CEO on AA production (Figure 3A). It was observed that the CEO improved the generation of AA (*p* < 0.05). Moreover, the additive amount of CEO at 0.5% and 1% (*v*/*v*) was more conducive to AA production than the additive amount of CEO at 3% (*v*/*v*). The Asn/Glc model system was heated for 10 min with 0%, 0.5%, 1%, or 3% (*v*/*v*) CEO, containing 21 mg/L, 54 mg/L, 34 mg/L, and 33 mg/L AA, respectively. No significant difference in AA concentration was observed between the Asn/Glc and the Asn/Glc/3% CEO model system when the heating time increased from 15 min to 25 min. The complex impact of CEO on AA production might be attributed to the carbonyl and phenolic compounds in chive. The carbonyl compounds in chive increase the production of AA [38,39]. On the other hand, the phenolic compounds in chive such as gallic acid and catechin have been reported to inhibit AA generation [40,41,42].

Ginger (*Zingiber officinale* Roscoe) is a spice used in Asian food for more than 3000 years due to its pungency, aroma, and nutrients [43]. Different concentrations of GEO were added to the Asn/Glc model systems to investigate the effect of GEO on AA production. After heating for 5 min, the addition of GEO showed no significant effect on AA production (Figure 3B). Subsequently, the concentration of AA increased rapidly in both the Asn/Glc and the Asn/Glc/GEO model systems with the reaction time prolonged from 5 min to 15 min. The AA concentrations demonstrated a downward trend with the reaction time increased to 25 min in the Asn/Glc and the Asn/Glc/GEO model systems containing 0.5% (*v*/*v*) or 1% (*v*/*v*) of GEO. The concentrations of AA increased continuously in the Asn/Glc/GEO model systems at the GEO additive amount of 0.5% (*v*/*v*). A similar increase in AA following ginger addition was found in pumpkin crisps [44]. On the contrary, the soaking of French fries in a ginger extract inhibited the production of AA [45]. In our previous study, we found that citral accounted for 89.89% of all the aldehydes in GEO [46]. Citral not only reacted with Asn to produce AA but also promoted the reaction between Asn and Glc to generate AA. Thus, citral was proposed as the major product which enhanced AA production.

Pepper (*Zanthoxylum bungeanum*) is one of the important spices, which causes a tingly, numb sensation on the lips. Linalool, limonene, and sabinene are reported as the major components of PEO [47]. Zeng et al. reported that pepper showed inhibitory effects on heterocyclic amine production in meats [48]. Nevertheless, the PEO enhanced the production of AA (Figure 3C) (*p* < 0.05). There was a dose–response relationship between the PEO level and the generation of AA. The level of AA increased during the whole heating process under all tested concentrations. After heating for 25 min, the concentration of AA was 55 mg/L, 63 mg/L, and 68 mg/L in the Asn/Glc/PEO model system at 0.5%, 1%, and 3% (*v*/*v*) PEO, respectively.

## 4. Conclusions

The addition of seasonings and spice essential oils showed complicated influences on AA production. The generation of AA was enhanced when low levels of soy sauce (1% and 3% *v*/*v*) and rice vinegar (1% *v*/*v*) were added. On the contrary, the addition of high levels of soy sauce (5% *v*/*v*) and rice vinegar (3% and 5% *v*/*v*) showed inhibitory effects on AA production. The addition of CEO, GEO, and PEO promoted the production of AA. Moreover, there was a dose–response relationship between the level of spice essential oils and the generation of AA. This study highlights the importance of seasonings and spices in AA production in food preparation. Further study should investigate the influence of seasonings and spices on AA generation in real foods. Moreover, the reaction mechanism of AA production in the presence of spice essential oils should also be studied. A technique to control AA production during food preparation without affecting the generation of flavor and aroma should be developed.

## Figures and Tables

**Figure 1 foods-11-03967-f001:**
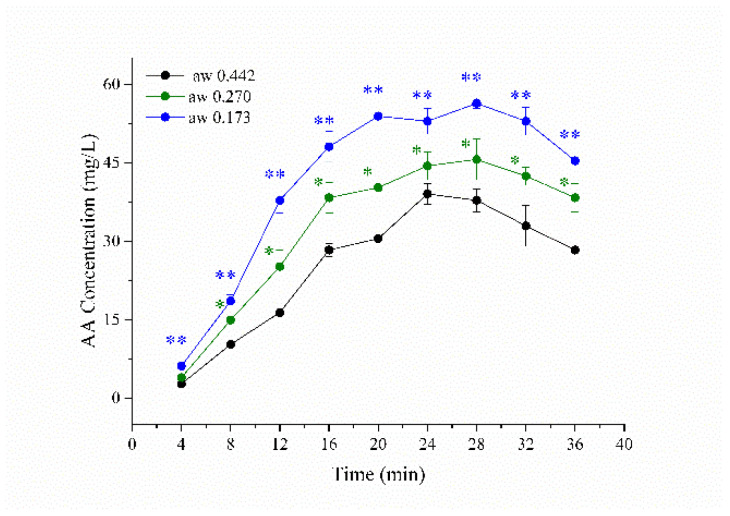
Effect of a_w_ on acrylamide (AA) production in the asparagine (Asn)/glucose (Glc) model systems heated at 180 °C for 25 min. Asterisks (*) denote a significant difference between the Asn/Glc model system at a_w_ 0.442 and the Asn/Glc model system at a_w_ 0.270. Two asterisks (**) denote a significant difference between the Asn/Glc model system at a_w_ 0.442 and the Asn/Glc model system at a_w_ 0.173.

**Figure 2 foods-11-03967-f002:**
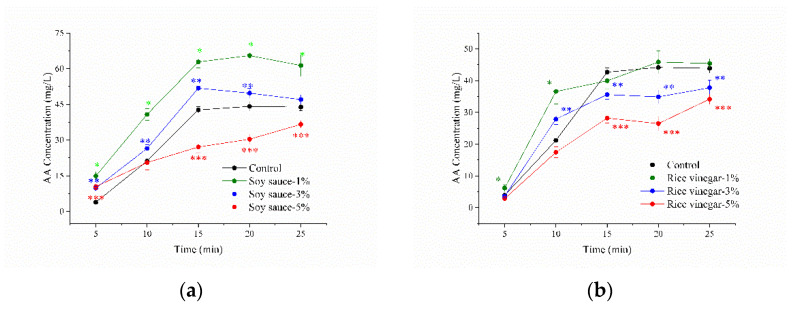
Effect of seasonings on AA production in the Asn/Glc/seasoning model systems heated at 180 °C for 25 min: (**a**) soy sauce; (**b**) rice vinegar. Asterisks (*), two asterisks (**), and three asterisks (***) denote significant differences between the control and the Asn/Glc/seasoning model systems at the additive amount of 1%, 3%, and 5% (*v*/*v*), respectively. The Asn/Glc model system is the control.

**Figure 3 foods-11-03967-f003:**
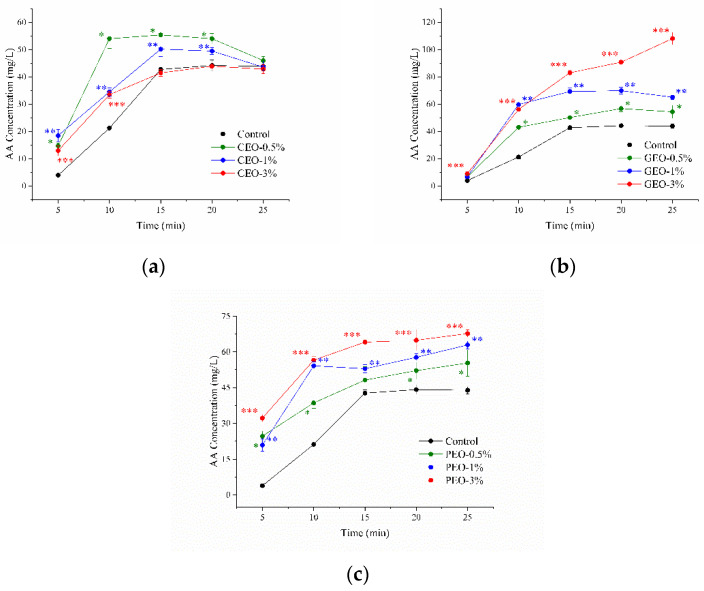
Effect of spice essential oils on the production of AA in the Asn/Glc/spice essential oil model systems heated at 180 °C for 25 min: (**a**) chive essential oil (CEO); (**b**) ginger essential oil (GEO); (**c**) pepper essential oil (PEO). Asterisks (*), two asterisks (**), and three asterisks (***) denote significant differences between the control and the Asn/Glc/spice essential oil model systems at the additive amount of 0.5%, 1%, and 3% (*v*/*v*), respectively. The Asn/Glc model system is the control.

**Table 1 foods-11-03967-t001:** Preparation of the model systems ^a^.

Model Systems	Diethylene Glycol (mL)	Deionized Water (mL)	a_w_
Model 1	4.75	0.25	0.173
Model 2	4.50	0.50	0.270
Model 3	4.25	0.75	0.442

^a^ All the model systems were 5 mL and contained 0.015 g asparagine (Asn) and 0.018 g glucose (Glc).

## Data Availability

All generated data are included in this article.

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
