# Peer review of "Influence of Seasonings and Spice Essential Oils on Acrylamide Production in a Low Moisture Model System"

_foods, 2022, doi:10.3390/foods11243967_

Round 1
Reviewer 1 Report
The presented study entitled “Influence of seasonings and spice essential oils on acrylamide formation in low moisture model systems” focuses on the production of AA in a controlled model system studying different concentrations of additives (three EOs and two seasonings). In general, the study is well written and structured. However, the results were not statistically processed. The paper only reflects the AA results obtained with the standard deviation. The authors should better to evaluate the real applications to reduce AA during food manufacturing processes, according to the main highlights concluded from the study.
The following minor and major suggestions should be considered by the authors to improve this contribution:
- Title: In my opinion, the title of the paper could be revised, because only one low moisture model system was studied
- Abstract: Please, to insert numerical results with significant difference in the abstract.
- Abstract: “Acrylamide (AA) is a typical contaminant produced in ready-to-eat foods and dishes”. This sentence is confusing; according to the sentence, AA could be produced in not processed foods.
- Line 34: Please, modify this sentence: “…in various foods and assess the intake of AA…” by “…in various foods and to evaluate the intake of AA…”
- Line 40: Please, complete this sentence as follow “…dietary source of AA was different in Asian countries”.
- Line 48: Please, change “industrialized dishes” by “prepared dishes”.
- Line 51: Please, modify this sentence as follow “…the seasonings and spices in the cooking process could also affect AA production”.
- Line 52: Why the authors choose these seasonings and these essential oils for the study? It would be interesting to include in the introduction section, the reasons of selecting these products for check AA production.
- Line 56: Please, translate this sentence to conclusions section.
- Lines 63-64: In my opinion, due to the importance of the essential oils used and the seasonings used, it is necessary provide a complete information, principally chemical characterization, in materials and methods section.
- Line 68: How were measured the water activities of the model systems? I did not find any analysis method in the text.
- Line 69: Which apparatus was used to heating the tubes at 180 ºC and controlling temperature?
- Lines 104-105: If the higher AA value was obtained with aw 0.173, why was selected the aw 0.27 model system concentration for the experiments?
- Line 94: In the results and discussion section, it is not reported if statistical testing was performed for the evaluation of AA production and therefore if the difference in AA content between treatments was significant.
- Lines 136-137: This sentence is rather inaccurate, because rice vinegar is produced from rice wine. The sugars are principally converted to ethanol due to alcoholic fermentation, and in another biochemical process this alcohol is transformed into acetic acid and others compounds by acetic bacteria.
- Line 138: If pH is an important factor to AA production, why have the authors not provided the pH results of the assays?
- Line 144: In figure 2 caption, change “(b), Vinegar.” by “(b) rice vinegar.”
- Line 148: Please, change “foramtion” by “production”, and to replace through whole the text.
- Line 172: The content of citral in GEO should be provided to compare with previous studies as the cited reference [42].
- Line 172: Please, change “Cit” by “Citral”.
- Lines 180-181: This sentence is unclear and should be rewritten. Please, consider the following correction to the sentence: “…the concentrations of AA were 55 mg/L, 63 mg/L, 34 mg/L, and 68 mg/L in the Asn/Glc/PEO model system at control, 0.5%, 1%, and 3% (v/v) PEO, respectively.”
- Figures 2 and 3: Please, change “CK” by “control”
- References: Please, the format of references should be standardized according to the requirement of FOODS journal.
Reviewer 2 Report
The manuscript is well presented and organized. The authors present the effect of different seasonings and spices on the formation of Acrylamide in food. The described experiments were properly planned and executed using a simulation system. The results were nicely presented and explained with comparison to previous studies. The study should be applied also on real samples as a further research.
Author Response
Thank you for the time and helpful suggestions for our manuscript. We plan to investigate the effect of seasonings and spices on AA generation in meatballs in our further study.
Reviewer 3 Report
Researchers tested the acrylamide formation in a system that consists, 2 seasonings and 3 spice essential oils. They found interesting results, that could help the food producers to inhibit the acrylamide formation. But the manuscript needs some major changes.
· Please pay more attention to the typos. For example, line 51 the word generation has a typo.
· The sentences at the last paragraph of the introduction needs to be reviewed.
· The introduction needs some information why the tested seasonings and the spice essential oils were chosen to be tested. I can understand the seasonings but still an explanation needs.
· What is the difference of this study from the previous ones? What is the unique part of this study should be given.
· Figures 2 and 3. What is CK? It is given in the figures, but I cannot see any explanation of what it is?
· The researchers did not try the application of soy sauce, vinegar, and the seasoning and spices essential oils in the real food systems, are they thinking the results will be similar with the real food systems? If they cannot test it at this point, at least they should mention it in the conclusion to provide an idea for the future researchers.
· So, what is your suggestion to the food producers?
· Do you think using both soy sauce and vinegar will increase the acrylamide formation after even after 5% addition?
Round 2
Reviewer 1 Report
In general the manuscript has been improved. The authors have adressed concerns in the rebuttal letter, but some of the explanations have to be added to the manuscript so the readers gets this important information: points 13, 16, and 19.
Author Response
Thank you for your comments. The points 13, 16, and 19 were added in the revised manuscript. All revisions of the manuscript were marked up using the “Track Changes” function.
Reviewer 3 Report
Authors have modified the manuscript according to my comments. A fine spell check may require.
Author Response
Thank you for your comments. The manuscript was checked carefully. All revisions of the manuscript were marked up using the “Track Changes” function.